# Addressing the Rehabilitation Needs of Women Experiencing Infertility in Ethiopia: Time for Action

**DOI:** 10.3390/ijerph21040475

**Published:** 2024-04-13

**Authors:** Bilen Mekonnen Araya, Maria P. Velez, Kassahun Alemu Gelaye, Silke Dyer, Heather M. Aldersey

**Affiliations:** 1Department of Rehabilitation Science, School of Rehabilitation Therapy, Queen’s University, 31 George St., Kingston, ON K7L 3N6, Canada; 2Department of Clinical Midwifery, School of Midwifery, University of Gondar, Gondar P.O. Box 196, Ethiopia; 3Department of Obstetrics and Gynaecology, Queen’s University, Kingston, ON K7L 2V7, Canada; maria.velez@queensu.ca; 4Department of Epidemiology and Biostatistics, Institute of Public Health, College of Medicine and Health Science, University of Gondar, Gondar P.O. Box 196, Ethiopia; kassalemu@gmail.com; 5Department of Obstetrics and Gynaecology, Groote Schuur Hospital and Faculty of Health Sciences, University of Cape Town, Cape Town 7701, South Africa; silke.dyer@uct.ac.za; 6School of Rehabilitation Therapy, Queen’s University, 31 George St., Kingston, ON K7L 3N6, Canada; hma@queensu.ca

**Keywords:** infertility, disability, rehabilitation, Ethiopia, religiosity, primary healthcare, reproductive health, Africa

## Abstract

The psychological, social, and financial disabilities caused by infertility are significant for women, particularly those living in low- and middle-income countries such as Ethiopia. Although rehabilitation can be an important form of support for such women, infertility is frequently overlooked as a disability or potential target of rehabilitation interventions. This study aimed to determine what rehabilitation-related services and supports are available for women experiencing infertility in Ethiopia. We used an Interpretive Description design. We purposefully selected fourteen rehabilitation, medical, and policy service providers from diverse institutions across three geographical locations. We used semi-structured questions during our in-person and telephone interviews. The data were analyzed using reflexive thematic analysis with the assistance of NVivo. We identified five main themes, including (a) policies related to infertility, (b) the concept that disabilities are physically visible fails to recognize infertility, (c) the need for rehabilitation services for women with infertility, (d) the importance of wellness services for women experiencing infertility, and (e) the role of religion in rehabilitation services. In conclusion, it is essential to strengthen the policies around infertility, incorporate rehabilitation services in fertility care, and view infertility as a disabling condition for women who experience it in Ethiopia.

## 1. Introduction

Approximately one in six people experience infertility at some stage in their lives [1,2]. Infertility—involuntary childlessness—is a disease or an impairment of a person’s capacity to reproduce characterized by the failure to achieve a clinical pregnancy after 12 months of regular unprotected sexual intercourse [3]. Infertility can occur regardless of whether one has had a prior birth (secondary infertility) or has not (primary infertility) [4]. An estimated 48 million couples globally and 186 million ever-married women in developing countries live with infertility [5,6,7,8,9]. The failure to achieve procreation indicates an impaired reproductive and endocrine system [10], generating a disability as an impairment of function [3]. Hence, infertility due to unsafe abortion and maternal sepsis is ranked by the WHO as the 5th highest serious global disability in low- and middle-income countries among women under the age of 60 [11].

The WHO International Classification of Functioning, Disability, and Health (ICF) consider factors related to infertility (i.e., oligospermia, azoospermia, subfertility, and sterility) as impairments of functions (i.e., female fertility, male fertility) [12]. In many instances, infertility leads to social exclusion and psychological trauma, amplifying the disablement process [3,9,11,13,14,15,16]. Infertility profoundly impacts an individual’s life, including their mental health, social life, marital status, and financial status [14,17,18,19,20,21,22,23]. However, little attention has been given to infertility as a disability [24,25,26,27,28]. Socially, recognizing infertility as a disability could lead to decreased stigma and psychological distress for those affected, which could lead to increased access to resources, such as insurance coverage for treatments and workplace accommodations [24,29]. Infertility can have a negative impact on the quality of life, especially in psychological and social aspects, of couples [30,31,32]. Studies show that childless women have poorer physical and mental health and social functioning compared to mothers [33,34]. Therefore, effective management of infertility is crucial for reproductive, psychological, and social health.

To endure the different consequences of infertility, people living with infertility need multidirectional and multidisciplinary support. Research indicates that, in addition to standard medical treatments, couples struggling with infertility require various services, such as professional psychosocial care, interventions focusing on the couple, and approaches prioritizing the patient’s needs [35]. Such patient-centered fertility care has significant clinical benefits, such as improving the client’s quality of life and emotional health and decreasing distress [30,31,32]. Additionally, religiosity can play a significant role in the lives of people experiencing infertility and other disabilities, as it relates to their coping and resilience, and can be part of the holistic service [36,37,38,39,40]. Religiosity is the degree to which someone is involved in organized religious activity (religious practice), the degree to which their religion influences their behavior (religious influence), and the degree to which a person feels hope in a religious sense (religious hope) [36].

Rehabilitation is crucial for achieving and maintaining optimal functional outcomes for individuals with health conditions. Rehabilitation is a set of interventions that aim to reduce disability and enhance functioning in the person’s interactions with their environment [41,42]. Rehabilitation contributes to achieving Sustainable Development Goal (SDG) 3—Good Health and Wellbeing—as it promotes healthy lives and well-being for all, regardless of age. Rehabilitation is not just for those with physical disabilities [43]. Instead, rehabilitation is an essential part of healthcare for those with acute or chronic health conditions, impairments, or injuries that limit functioning [44,45,46]. The underlying causes of infertility are equally male or female factors. However, in societies such as Ethiopia, where a woman’s worth and identity are closely linked to her ability to have children, women tend to bear the brunt of the psychosocial burden of infertility [20,25,47,48,49,50,51,52]. Women experiencing infertility could benefit significantly from rehabilitation services in various ways. These services can provide mental health support, address physical health concerns, offer education and decision-making support, promote holistic wellness, and facilitate peer support and community engagement.

The Ethiopian Reproductive Health strategic plan [53] and Family Planning guideline [54] include infertility as one of the deliverable services. However, fertility centers are limited in number and accessibility in Ethiopia, and rehabilitation specialists to support individuals with infertility are absent. Limited research exists on the perspectives of health service providers and policymakers in Ethiopia regarding rehabilitation and supportive services for women experiencing infertility. Hence, this study aimed to determine what rehabilitation-related services and supports are available for women experiencing infertility in Ethiopia. We explored the experiences of service providers approaching specific manifestations of infertility and their potential role in offering rehabilitation services to women with infertility—a population currently unserved by rehabilitation professionals. We speculate that examining infertility from a rehabilitation perspective can improve women’s quality of life by enhancing functioning and mitigating the emotional, physical, and mental challenges associated with infertility.

## 2. Materials and Methods

### 2.1. Study Design

This study uses an Interpretive Description (ID) design [55]. ID applies qualitative inquiry across health professions to capture clinical experience and inform practice, advancing evidence-based knowledge [56,57]. ID can inform practice by identifying how service providers can better attend to the wellness of women with infertility [57]. ID also promotes the use of the researchers’ disciplinary orientation [58]. In this case, BA has a background in midwifery and rehabilitation, MV and SD specialize in infertility and endocrinology, and HA’s expertise lies in disability and rehabilitation areas. Our respective disciplines guide our interpretation of the findings.

### 2.2. Study Population

Participants included social workers, psychologists, a Community-Based Rehabilitation (CBR) worker, an occupational therapist, clinicians, and policymakers. We included service providers working with persons with disabilities that supported them psychologically and socially, health care providers serving women experiencing infertility, and officials working on infertility at the Ministry of Health (MoH) level.

### 2.3. Study Setting and Context

The study was conducted in three cities in Ethiopia: Addis Ababa, Gondar, and Bahir Dar. Ethiopia had an estimated 120 million people in 2021 [59]. The institutions in which the service providers worked included trauma centers, specialized hospitals, rehabilitation centers and the MoH. Rehabilitation services for individuals experiencing infertility are unavailable in Ethiopia, as disability- and rehabilitation-related services are in their infancy. Nevertheless, people with physical and mental disabilities share some similar psychological and social manifestations as women with infertility. With that in mind, we interviewed participants on what they did for persons with disabilities for their psychological, social, daily activities, and spiritual well-being. Similarly, we held a speculative discussion on what the service providers could do, professionally, for women with infertility in the future. For the other healthcare providers, we focused on what they were doing to support women with infertility who have psychosocial problems and what more was required. With the MoH officials, we discussed the policies and guidelines concerning infertility and what more could be done.

### 2.4. Sampling Method and Selection Criteria

We purposefully included 14 individuals representing diversity in profession, gender, institution, and geographical location. Most participants were from Addis Ababa, since many healthcare institutions specializing in rehabilitation and infertility, including the MoH, are located there. The service providers were recommended by professionals working in disability- and rehabilitation-related projects. From the list of recommendations in each location and setting, we purposefully selected the participants offering psychological, social, and community-based services to persons with disabilities. We also added medical service providers and policy makers directly working on infertility. ID focuses on a deeper understanding of the participant’s perspective while recognizing the variation in perceptions rather than data saturation [55,60,61]. Participant recruitment was stopped using the principles of information power [62] by considering the study aim, sample specificity, and dialogue with participants.

### 2.5. Data Collection

The first author, BA, did eleven in-person and three telephone semi-structured open-ended interviews that lasted 30–60 min. She engaged the participants in a speculative thinking process by mentioning the manifestations of infertility that we identified in a previous study of women with infertility in Ethiopia [63]. “Speculative approaches are aimed at envisioning or crafting futures or conditions which may not yet currently exist, to provoke new ways of thinking and to bring particular ideas or issues into focus” [64]. At first, she did not mention that the manifestations were experienced by women with infertility specifically. In the information sheet and consent form, we only mentioned that the study was about the services they offer clients with psychological, emotional, occupational, and social problems. Such incomplete disclosure deception was needed to avoid or minimize the potential bias that may arise from the service providers, who may have a lack of knowledge or misconceptions about persons who experience infertility. Halfway through the interview, we included a question stating that these manifestations were taken from women diagnosed with infertility, and we inquired what specific services the service providers could offer to this population. Some of the questions we used were:What support do you offer people who experience psychological distress secondary to their disabilities?What support do you offer people who experience social isolation and stigmatization secondary to their disabilities?How do you see the role of spirituality in the services you provide?How can women with infertility be included in the current rehabilitation services in Ethiopia?

### 2.6. Data Analysis

ID allows a researcher to use data analysis methods that encourage engagement in data collection, broadly coding pieces of data, looking for patterns, and making sense of linkages in the patterns such as grounded theory and thematic analysis [55]. For this reason, we chose Reflexive Thematic Analysis (RTA) [65]. RTA is a qualitative data analysis approach that helps researchers identify patterns or themes in the data. It is flexible and emphasizes the researcher’s active role in knowledge production [65,66]. The interview was transcribed and analyzed in Amharic to minimize meaning loss during translation to English [67]. We further maintained the Amharic terms to minimize meaning loss for metaphors and sociolinguistic phrases [68] when we considered that the translated English would not capture the meaning. We translated the final themes into English during the write-up.

## 3. Results

Table 1 describes the characteristics of the 14 service providers included in the study. Most were from Addis Ababa, eight were male, and nine had more than five years of service in their field.

### 3.1. Background for the Services That the Study Participants Provided

The service providers delivered care to individuals with physical and mental disabilities, acute conditions such as the recent experience of a stillbirth, or chronic conditions such as cancer. They also supported families of children with physical, mental, and intellectual disabilities. The primary services they provided focused on psychosocial support, community awareness, referral, and linkage. They provided tailored counseling services, including cognitive behavioral therapy and motivational interventions to address specific needs such as depression, anxiety, and suicidal thoughts. Additionally, they offered coping and resilience treatments to help clients develop the necessary skills and strategies to manage difficult emotions and situations. They also worked on community education and awareness-promoting services to minimize stigmas related to disabilities. The providers also referred clients to different services based on their needs, including psychiatrists, social affair offices, and financial support organizations.

Below are the themes that arose from the interviews:

### 3.2. All Sizzle and no Steak—The Policy around Infertility

In this theme, participants discussed the guidelines and policies on infertility. During our conversation with the two officers from the Ethiopian MoH, we discussed the Ministry’s approach to infertility and potential areas for improvement were identified. The participants recognized that, although there was a policy on infertility, infertility did not receive sufficient attention, resulting in a significant policy implementation gap. Participant 14 stated, “*infertility is not an activity the ministry is focusing on like family planning. But infertility needs a separate focus and not just be included in FP*”. Participants identified several potential areas for improvement. According to them, infertility is mentioned in family planning and reproductive health guidelines, but with limited description and no elaboration on psychosocial support. The Ministry prioritized other areas due to the high fertility rate and pressing healthcare needs of the country, which means that infertility is often overlooked. However, the participants emphasized that infertility is a public health issue and not just a couple’s problem, and it needs to be addressed urgently. Participant 10 said, “*The disease burden of infertility is high in our community, especially the psychological problem. But I cannot say we are fully addressing the issue*”.

Additionally, most of the ministry’s projects were funded by non-governmental partners with specific interests and limited flexibility to address other issues, such as infertility. Participant 14 said, “*Infertility is not demanded and prioritized by partners*”. Consequently, fertility care is perceived as a “luxury” in Ethiopia, with no partner organizations actively working on the issue. The comment from participant 14 illustrates the above points: “*We have more emergent issues to focus on and have low resources … Infertility is not an issue many people suffer from in Ethiopia, and we have a high fertility rate*”. During their regular health institution supervisions, participants observed that infertility in primary healthcare institutions was a neglected issue. However, despite the perceived minimal effort put in by the MoH regarding infertility, participants indicated they were pleasantly surprised by the proactive measures taken by tertiary hospitals to address the issue. A participant indicated the following on how to address infertility.

“*According to the World Health Organization, health is complete physical, mental, and social well-being. Infertility can impact mental health, which is included in the non-communicable disease policy [in Ethiopia]. So, infertility can be included under mental health, and we just need to give it more attention*”(Participant 6)

### 3.3. Hidden in Plain Sight—The Narrow Definition of Disability as ‘Visible’ Limits Understanding of Infertility

When we initially approached participants, we intentionally did not disclose the specific population of women experiencing infertility. Our objective was to encourage participants to explore the diverse challenges that persons with disabilities face and subsequently demonstrate the similarities between the experiences of women with infertility and those with other disabilities. In this theme, we discuss how service providers view infertility from a disability perspective.

During our discussion, participants had varying opinions on whether infertility should be classified as a disability or not. Some expressed uncertainty, while others believed it could lead to a psychosocial disability and should be considered an invisible disability.

“*Infertility is a life-long issue. Subfertility could be considered as temporary, but infertility is permanent. One might get treatment but will not return to being fertile. So, like any other disabled person who has lost a leg or a hand, to some extent they [infertile people] are disabled. The word ‘disabled’ might not be great but it is better to see it that way if we want to think about rehabilitation*”(Participant 13)

Those who viewed infertility as a disability had a comprehensive understanding of the definition of disability and were aware of the psychological, social, and financial impacts of infertility. However, among some participants, there were differing views on whether infertility should be considered a disability since it can be treated. Participant 2 affirmed,

“*How we define disability is when there is something to see. For example, a blind person cannot do things a non-blind could do. The same with a physically disabled person. These people can be supported but cannot be cured. [Since we cannot see infertility, it is not a disability and since infertility may be treated, it is also not a disability]*”

The concept of infertility being treated was brought by participant 12 as well,

“*Infertility has its own cause like tubal blockage, azoospermia, and hormonal imbalance. The causes can be treated, and everything will be ok, or they can do IVF or hormonal treatment or take medications. So, it can be easily treated, and I cannot see it as a disability.*”

The Amharic term for disability “አካል ጉዳተኝነት”, was mentioned by most participants, who felt that it did not accurately reflect the concept of disability. This is why some could not visualize infertility as a disability, since there are no visible physical signs to distinguish an individual as infertile or not. Participant 11 elaborates,

“*Infertility is invisible. For a woman to know whether she is fertile or infertile, she needs to go and see a doctor. Then they will do hormonal, blood test and will tell her if she is infertile. Or the woman needs to tell people her problem of unable to conceive after trying for so long. Else no one would know or understand, that makes it invisible. The fact that it is invisible would make it more traumatic because the surrounding people can’t see it.*”

One informant expressed that rather than debating whether infertility was a disability or not, emphasis should be placed on understanding the psychosocial impact of infertility. Participant 3 also added,

“*Being under disability might have its own advantage and disadvantage. But for infertility to occur, there are body parts that are not functioning well so that is a disability. So, for now, it might be better to include it under disability and in the future infertility could have its own independent policy.*”

Although there were varying opinions on whether infertility should be considered a disability or not, all participants agreed on the necessity of including the psychological and social effects of infertility. Additionally, they emphasized the significance of extending support and assistance to individuals experiencing infertility to help them overcome the obstacles they encounter.

### 3.4. Rehabilitation Service for Women with Infertility

This theme highlights the specific types of rehabilitation services that participants believed were needed to address infertility, as well as their perceptions of gaps in current services. By identifying existing services, we can work towards improving the quality of care for women affected by infertility in Ethiopia, ultimately leading to improved health outcomes and quality of life.

Participants acknowledged that women with infertility were not receiving the necessary care within the current Ethiopian rehabilitation system. According to participants, rehabilitation services in Ethiopia were not well-developed and demanded. However, most participants agreed that women with infertility deserve to have a place within the rehabilitation system. Participant 4 pointed out that “*these women do not have a place in the current rehabilitation system. We lack rehabilitation generally, let alone for specific conditions*”. Participants stressed that women experiencing infertility require a holistic approach. This approach would ensure that these women receive the necessary care and support to improve their overall well-being. To cater to the diverse needs of these women, the participants suggested the establishment of a multidisciplinary team within a fertility center. The team would comprise various professionals, including psychologists, social workers, nurses, and medical doctors, who would work together to provide women experiencing infertility with various services. Participant 11 stated,

“*A lot can be done for these women. They need social, financial, and psychological support. If they are showing symptoms of depression, they can get treatment, support, and recover to get back to their normal life. Things can be arranged for them to get advanced treatment and education. They need to be educated that it can be treated, and they didn’t bring it on themselves. Many programs can be developed for them.*”

Some participants indicated that, in their professional/clinical education, they were taught how to assist with community reintegration after a client’s discharge from institutional care; however, due to the limitations of the current healthcare system, they did not engage in reintegration support at the time of the interview.

In addition to the necessity for community reintegration services, participants indicated that psychosocial rehabilitation and financial support could be provided alongside medical treatment. The objective would be to address holistic well-being, including physical, emotional, and social factors. Participant 12 shared, “*it would have been great if we could screen those [infertile] women who need rehabilitation and link them to a rehabilitation service provider*”. By providing comprehensive care, the participants believed that women experiencing infertility would be better equipped to cope with its social, emotional, and psychological effects, leading to an improvement in their overall quality of life. To address the knowledge gap in rehabilitation services among community members and service providers, participants suggested implementing a disability-sensitive pedagogy that introduces elementary and high school students to fertility, disability, and rehabilitation concepts early on. Service providers suggested that better community awareness of rehabilitation services would encourage more requests and strengthen government efforts in the field.

### 3.5. Services for the Wellness of Women Experiencing Infertility

This theme captured the perspectives of the participants regarding the services they were providing and the potential ways to improve their services to better support women struggling with infertility. The conversation centered on compassionate care for those facing infertility and a strong commitment to improving their lives.

#### 3.5.1. Focus of Existing Services—The Overlooked Psychological Support

The medical providers interviewed for this study acknowledged that their primary focus was on the medical treatment of infertility, which was typically very exhausting and stressful for clients. While closely working with women undergoing treatments such as surgery, egg retrieval, and hormonal stimulation, providers observed numerous psychological issues among their clients.

“*After they start the treatment, they encounter psychological burden because it is exhaustive. They get daily injections, repeated ultrasounds and so on. All these things are done to get a child and not because they are sick. Doing all this without being sick is hard and creates a lot of frustration*”stated participant 13

Participants currently working in medical management of infertility believed that further investigation would uncover even more psychological challenges. Some of the observed manifestations mentioned by participants included loneliness, suicidal thoughts, depression, and anxiety. Providers shared that they ‘try’ to provide counseling, although they acknowledged they lack sufficient psychological training. “*The psychological toll of infertility is unrecognized, but we have many cases. We need a trained clinical psychologist here at the fertility center, I do not think we [medical doctors] are experts on this*”, acknowledged participant 11. The service providers working with these women all agreed that their psychological support was inadequate, and that the fertility center required a designated psychologist. One clinician admitted, “*We recommend them [the women] to see a psychiatrist, but we do not have a psychologist in the center or a referral linkage system*”. Aside from the psychological support, a participant commented that they actively seek institutions offering subsidies for treatment supplies—hormonal injections, intra-uterine insemination (IUI) catheters, egg extraction needle, embryo preserving containers—to assist women facing financial limitations. The service providers also highlighted the importance for patients to maintain close communication with friends and engage in meaningful work to alleviate feelings of loneliness.

#### 3.5.2. Novel Services for the Wellness of Women with Infertility

Within the context of infertility, service providers who treated clients beyond those who experience infertility discussed the possible actions that could be taken to help women who are dealing with infertility. These providers had made some recommendations in the following areas: providing psychosocial support, increasing community awareness, making infertility treatment more affordable, and establishing self-help groups.

Service providers believed that women who experienced psychological trauma due to infertility needed family counseling and psychological support to help them not give up and move forward. Participants suggested that women may benefit from redefining the meaning of life for themselves and developing new, meaningful habits to help them accept their infertility and create a fulfilling life. As elaborated by participant 9, “*These women have lost meaning in their lives. When a woman reaches a certain age, giving birth would give more meaning to her life*”.

To provide women facing infertility with effective psychological support, it was considered important to offer reassurance and help them improve their coping skills. This may involve sessions aimed at redefining the meaning of life by exploring adoption and emphasizing gratitude by comparing oneself to those in ‘worse’ situations. The participants also suggested that fertility centers should have service providers who specialize or could take additional training in infertility, such as social workers, psychologists, and clinicians. Likewise, a CBR worker and a social worker who worked with other persons with disabilities indicated they could benefit from training on infertility. This would enable them to better support women experiencing infertility in their work.

Participants also suggested raising awareness in the community about the burden of infertility and making infertility treatment accessible. They believed that the community lacked knowledge on the causes of infertility, who can be affected by it, the stigma surrounding it, and where to get treatment. To address this gap, collaborators such as community and religious leaders, social workers, occupational therapists, CBR workers, and psychologists were suggested. “*There are a lot of barriers to work in the community because the bigger focus in our country is medical treatment*”, argued participant 6. Various methods, such as using media, shared social and religious gatherings, and door-to-door campaigns, were suggested to educate the community about infertility. Participant 9 emphasized, “*Working in the community brings significant changes compared to working on the individual level*”. Participants agreed that community mobilization would result in more people seeking infertility treatment at an earlier stage. Additionally, the government should improve access to treatment for low-income households.

Another suggestion was to create a self-help group for women who were experiencing infertility. The goal of this group would be to provide a safe space where women could come together, share their experiences, and support each other. Participant 1 said, “*they need to support themselves by discussing with one another by scheduling a discussion session and laugh with each other. I would love to work on counseling them or peer support groups*”. Additionally, a professional psychotherapist could provide the members of the group with group therapy. Participant 11, who works with women struggling with infertility, shared the plans they had to assist these women: “*We are planning to create a reproductive fertility society to create awareness among higher officials in the ministry of health about the impacts of infertility*”. An additional idea was also suggested by participant 8,

“*The educational curriculum [at schools] needs to include how people can preserve and continue their fertility, things that could cause infertility (hereditary or environmental causes), and when is the best time to give birth. If such information is included in the curriculum, children would know about infertility from the very young age.*”

### 3.6. Respecting Clients’ Religious Autonomy—The Use of Religiosity in Rehabilitation Services

The service providers who were working with persons with disabilities and women facing infertility discussed religiosity and its impact on the services they provided in depth. The participants unanimously agreed that religiosity plays a significant role in improving treatment outcomes, increasing psychosocial wellbeing, and overall health of their clients.

Participants shared insights on how religious beliefs, practices, and values influence the decision-making process of clients regarding their medical and rehabilitation therapies. The providers reported that their clients practice various religious traditions, such as seeking healing through holy water, attending religious services at churches and mosques, and engaging in prayer as a form of worship.

“*Incorporating religiosity in our treatment is important in the Ethiopian context. Most people think something bad is happening to them because they did something wrong or a sin. But when you tell them not to give up on the creator they believe in and that they are chosen for something sacred, they might give in easily. Our people are believers so when you use the religious road, they will hear you*”highlighted participant 7

These practices were an essential part of their religious lives and provided the clients with a sense of comfort and peace. “*Religiosity is a very important concept, especially to bring psychological motivation among our clients*”, elaborated participant 9. Despite differing philosophies, providers would find ways to offer both medical interventions and advice regarding religious practices side by side in a respectful manner. Participant 12 reflected, “*When we counsel women with infertility, they feel relieved and comforted when we tell them to see things from God/Allah’s rules and they are doing all that they can, and everything is up to the creator*”. A participant also mentioned that religious leaders would send clients with different conditions to them for medical and rehabilitation services.

The medical service providers emphasized the significance of religious beliefs in promoting psychological healing and inner strength among women experiencing infertility. The women were encouraged to establish a strong connection with their faith to achieve peace, resilience, and hope. Religiosity was recognized as having a transformative power to foster emotional well-being and personal growth. Participant 2 indicated, “*I would encourage women to have a medical diagnosis and treatment side by side with their strong religious beliefs to help them cope with the challenges they face from the community*”.

## 4. Discussion

The aim of this study was to explore the services and supports available to meet the rehabilitation needs of women experiencing infertility in Ethiopia. To achieve this, we used a speculative approach to discuss the issue with service providers who work with women experiencing infertility, as well as those who did not. The study identified potential areas for improvement and suggested expanding psychosocial and financial rehabilitation services. The psychosocial and financial rehabilitation services currently on offer for persons with disabilities could be expanded to include women experiencing infertility. Strengthening implementation of policies around infertility is important to provide women with holistic, accessible services. Rehabilitation service providers, clinicians, and policymakers should, therefore, be mindful of the weight of infertility on women’s lives and strive to provide comprehensive services that address the physical, psychological, social, and financial challenges related to infertility.

It is evident that there is a policy regarding infertility, but it has limited scope and lacks emphasis, something which does not benefit its target population. The Reproductive Health (RH) and Family Planning (FP) guidelines in Ethiopia recognize the burden of infertility [53,54]. However, our research reveals that infertility prevention and care services are not given enough attention as important public health issues. Instead, the focus is mainly on family planning to prevent pregnancy. We believe the policy should facilitate or direct the support and services women experiencing infertility would eventually receive. Policies can ensure that women have access to comprehensive healthcare, fertility treatments, mental health support, and other necessary services. Similarly, infertility clinics should focus on both medical and psychosocial aspects of infertility [52]. However, our study showed that only tertiary hospitals worked to address infertility treatments, while primary healthcare centers and communities did not.

Failure to address infertility at the primary level could potentially double healthcare costs [69]. A study shows that incorporating fertility care in primary healthcare is possible and advisable for low-income countries [70]. We believe that, to reduce the burden and cost of infertility treatment, it is essential to focus on prevention and early detection of infertility at the community and primary healthcare level. Education about fertility from a young age can also help guide people in fertility decision making process and awareness [71]. Therefore, investing in the implementation of policies to prevent infertility, as well as in timely treatment and rehabilitation of couples is crucial for promoting health equity, especially in low-income countries [69,72,73] such as Ethiopia. Furthermore, it is important to eliminate the stigma surrounding fertility treatments and recognize them as a necessary option for all women who need it [74,75]. Better policies and financial mechanisms are needed to educate people, reduce healthcare costs, increase access to health insurance, and provide those in need with financial assistance [69,72,75,76].

Our study highlights the fact that the current psychological and social rehabilitation programs in Ethiopia for individuals with disabilities could potentially benefit women experiencing infertility. These supports, as mentioned by participants, could involve psychotherapy, cognitive behavioral therapy, and motivational interventions to assist women struggling with psychological disorders. The American Disability Act defines a person with a disability as “someone who has a physical or mental impairment that substantially limits one or more major life activities” [77]. Among other non-physical unexplained causes, impairments to the reproductive and endocrine systems are physical causes of infertility. Reproduction is considered a major life activity [78], and the inability to achieve it not only affects a person’s physical health but also has significant psychological and social impacts on their life [79].

The greater physical and psychological disability and a poorer quality of life among women with infertility has been previously documented [80]. Therefore, rehabilitation programs including biopsychosocial support could help empower women to improve their overall psychosocial and financial well-being. Psychosocial rehabilitation helps achieve long-term emotional stability through personalized interventions that foster resilience and aid the adaptation to challenges. Moreover, individual or group therapy has been found to offer clients the opportunity to explore feelings in depth, share with others, and internalize that they are not alone [73,75,81]. Psychosocial support plays a crucial role in addressing the emotional, behavioral, relational, social, and cultural aspects of infertility [82,83].

Another possible strategy identified in our study was the opportunity for social and community-based rehabilitation workers to help educate communities about the preventable causes of infertility, decrease stigma and isolation, identify individuals with fertility issues, and connect them with appropriate healthcare services for early intervention and education. It is proven fact that social support can moderate the relationship between emotional disorders and marital adjustment [17]. Additionally, referral and linkage systems for accessing better psychological, social, and financial support could be established for these women. It is worth noting that, although women attending fertility centers in Ethiopia were encouraged to seek mental health support, they were often left to navigate the system alone without adequate guidance or support. Likewise, other studies have shown that fertility education should be an integral part of women’s healthcare to prevent unforeseen infertility and its psychological and social consequences [84,85].

Another interesting aspect of our study was the importance of incorporating religiosity in infertility-, disability-, and rehabilitation-related services in Ethiopia. Comprehending the significance of religion and utilizing healthcare services are equally important for people experiencing infertility [86]. Religious practices can help alleviate stress and increase positive emotions, leading to greater life satisfaction [36,87]. A previous study in Ethiopia has identified that participating in religious activities was important for persons with disabilities [40]. The provision of compassionate care and the recognition of the significance of religious beliefs and practices in the lives of their clients is important in Ethiopia. Providers must incorporate religious practices and values in their services to ensure personalized and effective care for clients and their psychological well-being [88].

## 5. Limitations and Strengths

It is important to consider both the strengths and limitations of this study. As a strength, this study is innovative with respect to its objective to explore the rehabilitation needs of women with infertility in Ethiopia. Additionally, the wide range of service providers included, conducting the interview in the local language, and the primary author’s familiarity with the language culture are strengths of the study. On the other hand, our study has some limitations. A possible limitation is the speculative approach, as service providers were not initially aware that the focus was on women with infertility. We suspected that there could be some negative bias or preconceptions from service providers who were not directly working with women experiencing infertility due to misconceptions or lack of knowledge about infertility. However, during the data collection process, we did not observe such bias on the part of the service providers we interacted with. Instead, we saw empathy and professionalism which could be genuine or due to the social desirability effect. We consider that our decision not to fully disclose infertility at the start was appropriate. This is because infertility is not yet recognized as a disability or a rehabilitation-related issue in Ethiopia. By not revealing that the focus was on women with infertility, we were able to promote a better understanding of the challenges these women face in their daily lives among the providers. Another limitation was the need to translate the study material from Amharic into English and the risk of some loss of meaning; however, this was safeguarded by the fact that the first author was bilingual.

## 6. Conclusions

The study’s results highlight the significance of infertility as a public health concern, and the need to enhance policies regarding it. Although more studies may be needed to determine if infertility is a disability, the consequences are disabling. Rehabilitation services—psychological, social, and financial—need to be integrated into the structure of fertility treatment. Furthermore, service providers working with persons with disabilities could offer better support to women experiencing infertility. Moreover, we should not overlook the positive impact of including religiosity in infertility and rehabilitation-related services in Ethiopia. Policymakers, clinicians, and rehabilitation workers should collaborate to address the needs of women facing infertility.

## Figures and Tables

**Table 1 ijerph-21-00475-t001:** Participants’ demographics and roles.

No	Profession	Gender	Location	Role at the Time of the Interview	Years of Service
1	Social worker	F	Addis Ababa (AA)	Rehabilitation center supervisor, assisting with physiotherapy and counseling for families	18 years
2	Rehabilitation worker	F	AA	Physical rehabilitation, assisting persons with disabilities in the daily activities of living, providing referrals for mental illness support	9 Years
3	Social worker	M	AA	Supporting those in need of economic, social, and psychological support up to rehabilitation	15 years
4	Medical social worker	M	AA	Social work department head, counseling, mediation, referral	7 years
5	CBR worker	F	Gondar	Going into the community and counseling and creating awareness on disabilities. Teaching braille for the visually impaired, sign language for the deaf, physiotherapy, involved in the assessment and provision of mobility devices, activities of daily living for intellectual disabilities	17 years
6	Clinical psychologist	M	Gondar	Counseling for patients in different departments	5 years
7	Psychologist	F	Bahir Dar	Counseling and guidance for persons with disabilities	4 years
8	Educational psychology	F	Gondar	Guidance and counseling of students with disabilities in groups and peer mentorship	24 Years
9	Occupational Therapist	M	Gondar	Teaching, research and community service, activities of daily living for persons with disabilities	3 years
10	Reproductive Health expert	M	AA	Policy and guideline development on reproductive health and family planning	18 years
11	Obstetrics and Gynecologist, fellow in infertility and reproductive endocrinology	M	AA	Providing clinical support to people experiencing infertility	5 years and 7 months
12	Nurse	M	AA	Giving clinical support to people experiencing infertility	3 years
13	General Practitioner	M	AA	Giving clinical support to people experiencing infertility	2 years
14	General Practitioner	F	AA	Policy and guideline development on reproductive health and family planning	3 years

## Data Availability

Data are available by a reasonable request to the corresponding author.

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
