# Peer review of "Addressing the Rehabilitation Needs of Women Experiencing Infertility in Ethiopia: Time for Action"

_ijerph, 2024, doi:10.3390/ijerph21040475_

Round 1
Reviewer 1 Report
Comments and Suggestions for Authors
Dear authors
firstly, I would like to praise you for shining light on infertility issues in Ethiopia and suggest you continue on conducting similar research in the future. However, I have a few remarks. In the introduction I suggest incorporating the definition of infertility into the text and not in the footnote. The same can be done with terms like religiosity and so on. Also, in the discussion I would recommend making the study limitations a separate subsection.
Author Response
Thank you very much for your suggestions. We have edited the introduction and discussion sections as suggested. Please see the highlighted sections in the revised version of the manuscript.
Reviewer 2 Report
Comments and Suggestions for Authors
STRENGTHS
a) The foundational arguments are well elaborated upon.
SUGGESTIONS
a) Please expound on the social exclusion and psychological trauma aspects of infertility. This will add more depth to the foundational basis for this article. Thank you.
b) Please explain with more detail how infertility is a form of disability. Please bear in mind the budding scholar who is being introduced to this concept for the first time and needs more information and guidance on the subject. Thank you.
c) As this is a qualitative study it should be bountiful in verbatim responses from the respondents. At present, it looks like the verbatim responses are lacking. May I please invite the authors to be more generous with the verbatim responses to add more facets and depth to the paper? Thank you.
CONCERNS
a) As incomplete disclosure deception was part of the data gathering process, would the authors please explain how they received ethical approval from the authorizing institution? In gist, how did the authors legitimize and justify this choice to the ethical review board? Thank you.
b) Although the authors purposively chose the respondents, what steps did the authors take to decrease bias? What was the selection criteria used by the authors for the respondents? Did any potential respondent who was approached decline the invitation, and if so how did the authors manage this? More information is needed in this section. Thank you.
Reviewer 3 Report
Comments and Suggestions for Authors
First of all, I would like to thank you for the opportunity to review the article - Addressing the rehabilitation needs of women experiencing infertility in Ethiopia: Time for Action. That said, some considerations can be outlined:
First, this is a very relevant topic when it comes to women's reproductive health. In low-income countries, this fact is not considered a health and a social problem, but rather a personal problem that can stigmatize women who are unable to have children. Ultimately, misinformation leads to this being a problem, which is not always addressed scientifically, which leads to discrimination and exclusion of women from society.
At the beginning of the text, fertility is treated as a disease, but soon after it is mentioned that it is a disability. The author must make these two concepts clear and without any shadow of a doubt. It does not seem that infertility can be defined as a disability, as this would bring other dimensions to the discussion and could be dangerous, as it would associate women and men who cannot have children with disabilities! For example in the WHO definition Infertility is a disease of the male or female reproductive system defined by the failure to achieve a pregnancy after 12 months or more of regular unprotected sexual intercourse. Nothing in this definition says it is a disability. https://www.who.int/news-room/fact-sheets/detail/infertility
In the third paragraph it states that infertility requires rehabilitation actions and mentions the millennium goals, relating to health and well-being. As it is a disease, I don't know if we can talk about rehabilitating it, but rather treating people with these diseases. This discussion needs to be substantiated or removed from the text as infertility is not a disability and therefore cannot be rehabilitated. But as a disease it can be treated or not treated. In the same paragraph, the case of Ethiopia highlights. This specific case needs to be very well clarified and would require a separate paragraph, due to cultural and religious issues. On the other hand, it would be relevant if the introduction demonstrated the scale of the problem (in numbers) not only in Ethiopia, but in global terms. As well as revealing the WHO guidelines for acting in these cases: example medical treatments and psychosocial support, etc.
In the population under study, the professionals and entities that participated in the study are identified. However, it goes on to say that rehabilitation services for individuals with infertility were not available in Ethiopia, as disability and rehabilitation services are still in their infancy. There seems to be a conceptual problem here. But are there rehabilitation services for people with fertility? Perhaps he meant psychosocial support, since the questions he asks go in that direction (although there is one that focuses on the integration of women in rehabilitation services).
The results provide a general framework for the topic, putting other variables into perspective, not always suitable for analyzing the infertility situation. As mentioned that there are no specific services for this disease, the existing services are also not prepared to respond to solve the problem: some provide support in kind, others access to services, but none of these services help women to have access to healthcare, or reproductive health and infertility treatment! The services are very generalist services. Of course, as health professionals point out, there are other more important problems in Ethiopia.
In fact asking professionals, if infertility was a disability only increases the unknown of the topic, and its stigmatization instead of doing the opposite. Sometimes there seems to be a disruption in discourse, between health professionals and others. I ask if this disruption was not induced by the questions asked – the speculative approach they use can be dangerous.
The results should unequivocally demonstrate the patterns identified in the various professionals interviewed: for doctors it is the need for innovative services; for others, it is psychosocial support and still others who consider that there should be a holistic and integrated response and eventually others who consider it a disability that It has to do with cultural and religious issues. These results should be approached critically. Women who cannot have children are not disabled.
Allong the article authors assume that infertility is a disability and this is a big problem in this study, especially for an international audience. That said, my appreciation of the article is not to be published.
Round 2
Reviewer 2 Report
Comments and Suggestions for Authors
Thank you for your revised version of the article. Thank you for addressing the comments and the feedback I had given previously.
Author Response
We truly appreciate the effort you put into reviewing our work and improving it. Your help means a lot to us, and we are grateful for your time and dedication. Thank you very much for your valuable feedback.
Reviewer 3 Report
Comments and Suggestions for Authors
The article aims to determine what rehabilitation-related services and supports were available to women with infertility in Ethiopia.
First: as I mentioned in the previous review, infertility is a disease that results in the inability of men and women to have children. There are, however, authors, doctors, who argue that it could be considered a physical disability or genetic anomaly. You can use this author to defend this argument.
See this author https://www.ncbi.nlm.nih.gov/pmc/articles/PMC3395292/.
As this author assumes, if infertility is considered as a disability, interventions can be created to combat the stigma and taboo that exists in the case of fertility in most cultures. Couples who are unable to reproduce may be looked down upon due to social stigmatization.
However, just defending this thesis does not mean that there will be less social exclusion, perhaps even exclusion increases when infertility is associated with a disability. It can certainly be understood as a deficiency of physical organs, but assuming this fact can bring to light the dark side of disability (stigma, discrimination, exclusion). That is, between one thing and another, there seems to be no better choice between these two positions.
Second, in the article the authors assume infertility as a disability. To argue this thesis, they refer to the WHO disability report, where they say that infertility is classified by the WHO as the 5th most serious global disability among women under 60 years of age. When looking at this report, in English, it only has the word infertility referenced twice: the first refers to disabilities associated with preventable causes, such as unintentional injuries and infertility originating from unsafe abortions and maternal sepsis (page 296) and the second is on page 297 in the table presented on that page.
Nowhere in this report written in English does it mention that infertility is classified by the WHO as the 5th most serious global disability among women under 60 years of age, as is written in the article on the first page see https://iris.who.int /handle/10665/44575
I advise you to review this note, perhaps it is from another report that I am not familiar with.
Third, the word rehabilitation is used, but it is never clearly explained what rehabilitation means: whether they are medical support services for these women and men and or whether they are social services that have that name, or whether they are both. It is necessary to clarify to the audience what the author understands by rehabilitation: are these services that exist in the community to support women? These are not specific in this area, but they support and help women to overcome, from a psychosocial point of view, the fact of not having children, and are these necessary to support these people in the face of the discrimination they may suffer for cultural and religious reasons?
Fourth, although the study is relevant, there is a disruption here that has to do with the different responsibilities and training of the professionals involved. Because not everyone undergoes psychosocial “rehabilitation”. One suggestion is for the author to look at the results and focus on what they bring to the analysis. Clearly what these professionals do is provide biopsychosocial support to these women. So why not call it biopsychosocial support instead of rehabilitation? Rehabilitation implies recovering, redoing, something that doesn't work well... in this specific case what these professionals do is work to restore rights, capacity and social, cultural and religious position lost after knowing they are infertile.
Fifth, the questions asked by the authors to the participants focus on: 1. What support do you provide for people who experience psychological distress secondary to their disabilities? 2. What support do you provide for people who experience social isolation and stigmatization secondary to their disabilities? 3. How do you see the role of spirituality in the services you provide? 4. How can women with infertility be included in the current rehabilitation services in Ethiopia, only this last one concerns the topic, so the rest should not be considered as they focus on disabled people in general.
If you consider all the questions in the analysis, then the article must include other disabled people and not just those who are infertile, which implies changing the initial part of the article. Perhaps the authors intended to focus on the rehabilitation of disabled people including people who are infertile. Or just erase the first questions.
Finally, a last suggestion is to use the title “Services for infertile women in Ethiopia” instead of rehabilitation services, as this would perhaps be more in line with the study carried out, as not all interviewed professionals promote rehabilitation services, but before support, counselling, or clinical support.
I wish you good luck
